# Use of the AE Effect to Determine the Stresses State in AAC Masonry Walls under Compression

**DOI:** 10.3390/ma14133459

**Published:** 2021-06-22

**Authors:** Radosław Jasiński, Krzysztof Stebel, Paweł Kielan

**Affiliations:** 1Department of Building Structures and Laboratory, Faculty of Civil Engineering, Silesian University of Technology, 44-100 Gliwice, Poland; radoslaw.jasinski@polsl.pl; 2Department of Automatic Control and Robotics, Silesian University of Technology, 44-100 Gliwice, Poland; 3Department of Mechatronics, Silesian University of Technology, 44-100 Gliwice, Poland; pawel.kielan@polsl.pl

**Keywords:** masonry structures, autoclaved aerated concrete masonry units (AAC), compressive strength, minor-destructive (MDT) techniques, non-destructive (NDT) techniques, ultrasonic testing, acoustoelastic effect (AE), hydrostatic stresses, modeling, DIC technique

## Abstract

Safety and reliability of constructions operated are predicted using the known mechanical properties of materials and geometry of cross-sections, and also the known internal forces. The extensometry technique (electro-resistant tensometers, wire gauges, sensor systems) is a common method applied under laboratory conditions to determine the deformation state of a material. The construction sector rarely uses ultrasonic extensometry with the acoustoelastic (AE) method which is based on the relation between the direction of ultrasonic waves and the direction of normal stresses. It is generally used to identify stress states of machine or vehicles parts, mainly made of steel, characterized by high homogeneity and a lack of inherent internal defects. The AE effect was detected in autoclaved aerated concrete (AAC), which is usually used in masonry units. The acoustoelastic effect was used in the tests described to identify the complex stress state in masonry walls (masonry units) made of AAC. At first, the relationships were determined for mean hydrostatic stresses *P* and mean compressive stresses *σ*_3_ with relation to velocities of the longitudinal ultrasonic wave *c_p_*. These stresses were used to determine stresses *σ*_3_. The discrete approach was used which consists in analyzing single masonry units. Changes in velocity of longitudinal waves were identified at a test stand to control the stress states of an element tested by the digital image correlation (DIC) technique. The analyses involved density and the impact of moisture content of AAC. Then, the method was verified on nine walls subjected to axial compression and the model was validated with the FEM micromodel. It was demonstrated that mean compressive stresses *σ*_3_ and hydrostatic stresses, which were determined for the masonry using the method considered, could be determined even up to ca. 75% of failure stresses at the acceptable error level of 15%. Stresses *σ*_1_ parallel to bed joints were calculated using the known mean hydrostatic stresses and mean compressive stresses *σ*_3_.

## 1. Introduction

The ultrasonic technique [1,2,3,4] is used in spectroscopy, defectoscopy, evaluation tests, coagulation, dispergation, sonoluminescence cavitation, and chemical reactions. Ultrasounds can be also applied to crush, form hard media, bond, solder, wash, extract, and dry substances. Another important application of these ultrasonic techniques includes stress measurements in metal constructions. Ultrasonic methods of measuring stress use the acoustoelastic effect (AE), i.e., the correlation between the stress and the velocity of acoustic wave propagation.

The ultrasonic pulse velocity (UPV) is a method applied to cement (concrete) and ceramic materials. This method is used to determine a setting time, changes in the elasticity modulus, and to test the compressive strength (only with the applied minor non-destructive (MDT), technique) [5,6]. Besides the AE tests conducted on isotropic materials, the current experience and theoretical analyses of construction materials are related to the anisotropy effect on the wave propagation. The tests mainly include almost isotropic, moderately and strongly anisotropic metamaterials at shear strain and standard deformation. Reference [7] describes the quantitative effect of finite deformations with reference to their magnitude and load direction. Strain-induced instabilities cause negative increments in the phase velocity just as in the case of the isotropic materials. It was also demonstrated that shear strains did not change the velocity of longitudinal waves as the material volume was stable. On the other hand, the effect of high values of deformations on the propagation of acoustic waves in repetitive network materials was explored in the paper [8]. The deformations were found to significantly affect the frequency waves and the phase velocity. In particular, the phase velocity for the hexagonal network strongly decreased under finite compressive deformations. The effective density was shown to have an important impact on the dispersion relation and band diagrams under the application of incremental deformation over the lattice unit cell. Additionally, the theoretical analyses [9,10] are made on mechanical wave propagation in the infinite two-dimensional periodic lattices using Floquet-Bloch. Conclusions derived from the tests can be applied to research on orthotropic construction materials (composites, composite panels, homogeneous masonry structures, etc.). The paper [11] showed that the acoustoelastic effect (AE) [12] also occurs in autoclaved aerated concrete (AAC). The stress state in the wall made of AAC masonry units was determined on the basis of the conducted analyses. 

This paper describes the tests aimed at determining the complex state of stresses in masonry units made of autoclaved aerated concrete. As in [11], verification tests were performed on small parts of the wall subjected to axial compression. Their aim was to define empirical relationships of mean hydrostatic stresses *P*, and then normal stresses *σ*_1_ and *σ*_3_ in the wall, in which the AE effect was observed [12]. Taking into account the AAC vulnerability to moisture content [13] which deteriorates insulation and strength parameters, the analyses included both density and relative humidity of this material. This paper demonstrates a practical application of the AE effect in testing masonry structures which was described in previous works of the author [11,14,15]. The tests were divided into two stages. In the first stage of the tests, experiments were performed on 24 small cuboidal specimens (180 mm × 180 mm × 120 mm) of autoclaved aerated concrete with nominal densities of 400, 500, 600, and 700 kg/m^3^. The obtained results were used for determining the acoustoelastic constant *δ_P_* that showed the relationship between mean values of hydrostatic stress *P* and velocity of the longitudinal wave *c_p_*. 

Stage II included nine wall models [11] made of AAC masonry units which had a nominal density of 600 kg/m^3^. They were subjected to axial compression The velocity of ultrasonic wave *c_p_* in the masonry units was measured. The complex stress state in the wall was examined using the relationships established in stage I. Then, the linear elastic FEM models was applied to match the *P*–*c_p_* relationship. Knowing stress values *σ*_3_ determined in [11], the levels of normal stress *σ*_1_ could be determined.

## 2. Theoretical Bases of the AE Method

Stress in the material can affect velocity of the acoustic wave because of inhomogeneity and anisotropy. That effect was theoretically described for the first time in the paper [16], and the experimental verification was presented in the papers [17,18]. The static stress was found to have an impact on changes in the velocity values of the acoustic wave in the medium. This pattern has been known as the acoustoelastic (AE) effect [19,20].

This effect, whose theoretical background was described in the paper [12,21], specifies the relationship between stress and velocity of transverse wave propagation. As from then, this subject has significantly evolved [22,23,24]. The normal stresses have an impact on a change in the velocity of longitudinal and transverse waves (as in the elastooptic effect involving light waves), which is determined by the direction of wave propagation over the direction, the stress, and the wave polarization. Following the theory of solid deformation [17], the higher orders elasticity constants (neglected in the linear theory of elasticity) which describe the non-linear effects, should be taken account during the analysis of the AE effect. A sum of velocities in the tensionless state (*σ* = 0) and its change (an increment) as a result of the stress (strain) expresses the velocity of ultrasonic wave propagation.

In accordance with the Murnaghan theory [25], the function of free energy *W_s_* defined below [17,26], is described by the stress-deformation relationship
(1)$$W_{s}=\frac{1}{2}\left(\lambda +2\mu \right)I_{1}^{2}-2\mu I_{2}+\frac{1}{3}\left(l+2m\right)I_{1}^{3}-2mI_{1}I_{2}+nI_{3},$$
where: *λ*, *μ*–Lamé constants, *l*, *m*, *n*–elasticity constants of second and third order by Murnaghan, *I*_1_, *I*_2_, *I*_3_–deformation invariants. 

Following the principle of energy conservation, Hooke’s law can be given by
(2)$$\rho \delta W_{s}=\sigma_{ij}\frac{\partial \delta u_{i}}{\partial u_{j}},$$
where *δW* and *δu_i_* mean finite increments in the function of free energy and the displacement area, *ρ* is density after deformation (in the stressed body). This AE equation specifies the relationship between the static load and the elastic wave velocity under hydrostatic conditions (that is, under the hydrostatic stress *P*)–Figure 1
(3)$$c_{p}^{2}=\underset{c_{p0}^{2}}{\underset{︸}{\frac{\lambda +2\mu }{\rho_{0}}}}-\frac{P}{\rho_{0}\left(3\lambda +2\mu \right)}\left(6l+4m+7\lambda +10\mu \right),$$
(4)$$c_{t}^{2}=\underset{c_{t0}^{2}}{\underset{︸}{\frac{\mu }{\rho_{0}}}}-\frac{P}{\rho_{0}\left(3\lambda +2\mu \right)}\left(3m+0,5n+3\lambda +6\mu \right)\;,$$
where *c_p_* and *c_t_* are velocity of longitudinal and transverse waves respectively, and *ρ*_0_ body density in the tensionless state, *P*–hydrostatic stress defined as $P=\frac{1}{3}\left(\sigma_{1}+\sigma_{2}+\sigma_{3}\right)$.

The Equation (2) [27] can be used to determine the stress *P*. For that purpose velocity of the longitudinal and transverse waves is measured. The squared velocities of waves at uniaxial stress states are expressed by these equations
(5)$$V_{111}^{2}=c_{p0}^{2}-\frac{\sigma_{1}}{3K_{0}\rho_{0}}\left[\frac{\lambda +\mu }{\mu }\left(4\lambda +10\mu +4m\right)+\lambda +2l\right],$$
(6)$$V_{113}^{2}=c_{p0}^{2}+\frac{\sigma_{3}}{3K_{0}\rho_{0}}\left[\frac{2\lambda }{\mu }\left(\lambda +20\mu +m\right)-2l\right],$$
(7)$$V_{131}^{2}=c_{t0}^{2}-\frac{\sigma_{1}}{3K_{0}\rho_{0}}\left[4\lambda +4\mu +m+\frac{\lambda n}{4\mu }\right],$$
(8)$$V_{133}^{2}=c_{t0}^{2}-\frac{\sigma_{3}}{3K_{0}\rho_{0}}\left[\lambda +2\mu +m+\frac{\lambda n}{4\mu }\right],$$
(9)$$V_{132}^{2}=c_{t0}^{2}+\frac{\sigma_{2}}{3K_{0}\rho_{0}}\left[2\lambda -m+\frac{n}{2}\frac{\lambda }{2}\frac{n}{\mu }\right],$$
where $K_{0}=\frac{E}{3\left(1-2\nu \right)}=\frac{2\mu +3\lambda }{3}$, $c_{t0}=\sqrt{\frac{\mu }{\rho_{0}}}$, $c_{p0}=\sqrt{\frac{\lambda +2\mu }{\rho_{0}}}$.

The velocity of the ultrasonic wave (in the deformed material), elastic constants of the first (*λ*, *μ*), second and third order (*m*, *n*, *l*), whose detection is the most difficult in the tests, are used to determine the normal stresses in the material.

The theoretical background of the AE effect has been adequately proved. Also, the suitable equipment is employed to determine the elastic constants of the third order *l*, *m*, *n* for metal and plastic materials following the procedures presented in e.g., the papers [28,29,30,31]. Knowing the direction of the exerted load and a gradient of changes in the longitudinal or transverse wave velocity is required to examine the stress states with the NDT technique.

The proposed procedures can be easily applied to the laboratory tests, however, their use under the in-situ conditions can be troublesome. Hence, a relative increment in the longitudinal wave velocity [32] (knowing the Murnaghan coefficients is not required) is more favorable for practical applications and it can be obtained from the following relationship (based on the Equation (3))
(10)$$\begin{array}{l} c_{p}^{2}-c_{p0}^{2}=-P\frac{\left(6l+4m+7\lambda +10\mu \right)}{3\rho_{0}K_{0}}\to (c_{p}^{}-c_{p0}^{})(c_{p}^{}+c_{p0}^{})=-P\frac{\left(6l+4m+7\lambda +10\mu \right)}{3\rho_{0}K_{0}},\\ \mathrm{assuming}\;\mathrm{that}\;c_{p}+c_{p0}\approx 2c_{p0}\to (c_{p}^{}-c_{p0}^{})2c_{p0}^{},\\ \mathrm{the}\;\mathrm{following}\;\mathrm{was}\;\mathrm{obtained}\\ \;\;\;\;\frac{\left(c_{p}^{}-c_{p0}^{}\right)}{c_{p0}}=-P\frac{\left(6l+4m+7\lambda +10\mu \right)}{6\rho_{0}K_{0}c_{p0}^{2}},\\ \mathrm{taking}\;\mathrm{into}\;\mathrm{account}\;\mathrm{the}\;\mathrm{following}\;\mathrm{terms}\\ \;\;\;\;K_{0}=\frac{2\mu +3\lambda }{3};c_{p0}^{2}=\frac{\lambda +2\mu }{\rho_{0}},\\ \mathrm{finally},\;\mathrm{we}\;\mathrm{obtain}\\ \;\;\;\;\frac{\left(c_{p}^{}-c_{p0}^{}\right)}{c_{p0}}=-P\frac{\left(6l+4m+7\lambda +10\mu \right)}{2\left(2\mu +3\lambda \right)\left(\lambda +2\mu \right)}=P\delta_{P}. \end{array}$$
where *δ_P_* is the AE coefficient expressing the relationship between a relative increment in the longitudinal wave and the mean hydrostatic stresses.

The relative AE coefficient can be expressed as
(11)$$\begin{array}{l} \frac{\left(c_{p}^{}-c_{p0}^{}\right)}{c_{p0}}=\frac{P}{P_{\max}}\eta_{P}.\\ \; \end{array}$$
where *η_P_* is the relative AE coefficient expressing the relationship between a relative increment in the longitudinal wave and the relative mean of hydrostatic stresses.

The paper [11] defined values of the AE coefficients under the uniaxial compression in the form
(12)$$\frac{\left(c_{p}-c_{p0}\right)}{c_{p0}}=\beta_{113}\sigma_{3}.$$
(13)$$\frac{\left(c_{p}-c_{p0}\right)}{c_{p0}}=\gamma_{113}\frac{\sigma_{3}}{\sigma_{3\max}}.$$
where
(14)$$\beta_{113}=1.39\cdot 10^{-4}\rho -0.104,\;R^{2}\;=0.995,\;\gamma_{113}=1.72\cdot 10^{-4}\rho -0.206,\;R^{2}\;=0.923,$$
when $397\;\frac{\mathrm{kg}}{m^{3}}\leq \rho \leq 674\;\frac{\mathrm{kg}}{m^{3}}$.

## 3. Program of Own Research

Following the procedure described in the paper [11] the tests were divided into two stages. In the first stage of the tests, the biaxial compression was exerted until the failure of the specimens with dimensions of 180 mm × 180 mm × 120 mm. Velocity of the longitudinal wave was determined under different hydrostatic stress *P*. The tests were conducted in a test stand specially prepared to test the specimens and simultaneously control their deformations by the non-contact technique of Digital Image Correlation DIC. The obtained results were the base to determine the linear correlations of the *c_p_*–*P* relationship. The test results for nine masonry models under axial compression, described in [11], were used in the stage II to determine at first mean hydrostatic stress, and then the normal stress *σ*_1_ which was parallel to the plane of bed joints. The test results for the complex state of stresses were compared with the results for the linear-elastic FEM models. Then, the method was validated.

## 4. Test Results

### 4.1. Stage I-Determination of Acoustoelastic Constant

#### 4.1.1. Physical and Mechanical Properties of Autoclaved Aerated Concrete AAC

The tests included four series of masonry units with a thickness within the range of 180–240 mm and different classes of density: 400, 500, 600, and 700 [33], which were the subject of tests presented in the paper [11]. Six cores with a diameter of 59 mm and a height of 120 mm were cut out from the masonry units. They were used to determine the fundamental properties of the test autoclaved aerated concrete (AAC). All the cores were dried until constant weight at a temperature of 105 ± 5 °C. The modulus of elasticity *E* and Poisson’s ratio *ν* were determined for the core specimens. Mean mechanical parameters obtained for all the tested types of masonry units are shown in Table 1. The results from testing density and compressive strength of the specimens 100 mm × 100 mm × 100 mm were taken from [14].

Apart from the core specimens of AAC masonry units, also 24 rectangular specimens having dimension of 180 mm × 180 mm × 120 mm were cut out and used in the stage I. To determine the correlation between mean hydrostatic stress *P* and ultrasonic velocity, all the specimens were air-dried at a temperature of 105 ± 5 °C for at least 36 h until constant weight. That way the impact of moisture content on AAC was eliminated [15,34]. Generally, moisture content tends to significantly reduce compressive strength and change velocity of the ultrasonic wave propagation [14]. 

#### 4.1.2. Test Stand and Procedure

The velocity of ultrasonic waves was determined by the method of transmission [11,35,36]. Velocity of ultrasonic waves in 180 mm × 180 mm × 120 mm specimens taken from the masonry units, was measured at the specially prepared test stand—Figure 2. The test stand for testing biaxial compression consisted of two vertical columns 1 made of a set of two channel profiles 120 with a length of 1000 mm and connected at the bottom with a spandrel beam 2 made of three I-beams 140 with a top spandrel beam 3 which was made of an I-beam 200 with a length of 1000 mm and reinforced with ribs. Inside dimensions between spandrel beams and the column were 820 mm in a vertical plane, and 810 mm in a horizontal plane. Openings with the spacing of 75 mm were made in vertical columns 1 and in the spandrel beam 3 to change its position. The hydraulic actuator 4 with an operating range of 500 kN was pin jointed to the top spandrel beam. A draw-wire displacement converter 5 of SWH-1-B-FK-01 type with the TRA50-SA1800WSC01 encoder (TWK-ELEKTRONIK GmbH, Düsseldorf, Germany) was attached to the side wall of the actuator. The hydraulic actuator was connected to the hydraulic power unit “A” (Zwick Roell Company Group, Ulm, Germany) with a pressurized pipeline, to which the pressure transmitter P30 was attached (WIKA SE & Co. KG, Klingenberg, Germany) 6. Its operating range was 0–1000 bar and the reading accuracy was 1 bar. The hydraulic actuator 7 with an operating range of 500 kN was sliding jointed to one vertical column. A draw-wire displacement converter 8 (SWH-1-B-FK-01 type with the TRA50-SA1800WSC01 encoder) was fixed to the actuator. The actuator was connected to the hydraulic power unit “B” (Hydac International GmbH, Sulzbach/Saar, Germany) with a pressurized pipeline, to which the pressure transmitter P30 was attached-9 The research model 10 was placed between Teflon washers 11 and steel plates 12 with ball joints. 

This test stand was a complex system designed and prepared by the authors [37]. This design is copyrighted [38]. The advanced control algorithms had to be applied as many non-linearities were present in the subsystems. These algorithms ensured the proper interactions between elements of the test stand. Due to the continuous improvement of these algorithms [39,40], the test stand performance is characterized by high repeatability as proper feedback is ensured among the following components of the system:Hydraulic systems “A” (Zwick Roell Company Group, Ulm, Germany) and “B” (Hydac International GmbH, Sulzbach/Saar, Germany),Electrical system: developed by authors’ of the testsPeripheral devices: the model P30 pressure transmitters (WIKA SE & Co. KG, Klingenberg, Germany), the draw-wire displacement converters of SWH-1-B-FK-01 type with the TRA50-SA1800WSC01 encoder (TWK-ELEKTRONIK GmbH, Düsseldorf, Germany), the Digital Image Correlation System ARAMIS 6M ((GOM GmbH, Braunschweig, Germany), the PUNDITLAB+ instrument for reading and recording ultrasonic waves (Proceq Europe, Schwerzenbach, Switzerland),The measurement and control interface: based on the NIcRIO 9022, NIcRIO 9056 controller (National Instruments, Austin, TX, USA),IT system: developed by the authors in the LABVIEW 2020 software (National Instruments, Austin, TX, USA) [41].

The block scheme in Figure 3 illustrates the interactions between individual elements of the system. The IT system with the hydraulic system generated stresses *σ*_1_ and *σ*_3_ of the same value and were used to read the ultrasonic wave path recorded with the PUNDIT LAB+ instrument. The ARAMIS 6M system was used to control deformations and observe crack images in individual specimens. Collecting data from different subsystem in one IT system ensured an additional option for the tests due to the time correlation of many data and their cause–effect relations. When different systems were combined, the set tasks were performed in a more effective way compared to individual subsystems [42,43,44,45].

The tests were conducted on the specimens which were dried to constant weight and which had relative humidity *w*/*w*_max_ = 0%. The tests included at least 6 specimens of the same density, and 24 specimens in total were tested (Figure 4a,c,d). The PUNDIT LAB+ instrument (Proceq SA, Schwerzenbach, Switzerland), which was integrated with the IT system of the test stand, was used to measure velocity of ultrasonic waves. The point measurements were taken with the exponential transducers the frequency 54 kHz (Figure 4e). The measurement accuracy of passing time of the ultrasonic wave was equal to ±0.1 μs. Each specimen was placed between the plates of the test stand using Teflon washers of 10 mm in thickness. Compressive stress *σ*_3_ was generated in the vertical direction. In the horizontal direction, in which the normal stress *σ*_1_ was generated, Teflon plates, and then steel sheet were placed on the lateral sides to generate loading. The measuring templates were placed to end face of each specimen (Figure 4b) in the next step. The passing time of the wave was measured with transducer which were in put (at 90°) into the openings of the measuring templates. Each time a distance was measured between the transducers with an accuracy of 1 mm. An increment in stress values could be uniform by controlling loads exerted in both vertical and horizontal directions by hydraulic actuators ‘A’ and ‘B’. Velocities of ultrasonic waves were read every 5 kN (for the specimens with nominal densities of 400 and 500 kg/m^3^) and every 10 kN (the specimens with nominal densities of 600 and 700 kg/m^3^). A view of the test stand in operation is shown in Figure 5.

#### 4.1.3. Test Results

There were not any models with damaged front face during the loading cycle. Prior to the failure crack was heard and noticeable cracks were observed on the surface. Noticeable cracks were also found on the specimen surfaces under loading that preceded the failure. Debonding of external surface of each test element was observed at failure. It revealed a type of the specimen damage with clearly truncated pyramids that were connected in the center of the specimen. Passing time of the ultrasonic wave using the transmission method was measured at 25 points of each specimen at the following stress values: 0, ~0.25*P*_max_, ~0.50*P*_max_, ~0.75*P*_max,_ *P*_max_. Examples of the obtained maps showing passing time of the wave are illustrated in Figure 6, Figure 7, Figure 8 and Figure 9.

Velocities of ultrasonic waves in all the specimens were significantly disturbed in the edge areas. Noticeably lower wave velocities were observed in these areas. The results referred to 16 points (as shown in the template–Figure 4a): A1–A5, B1, B5, C1, C5, D1, D5, E1, E5, and F1–F5. The observed disturbances were caused by the immediate vicinity of loaded edges of the specimens and the recorded wave reflection at the edge, and also by local damage to the material during the loading phase. The highest homogeneity of the results was found in central areas of each specimen at nine points B1–B3, C1–C3, D1–D3, and E1–E3. Table 2 presents the measurement results for the ultrasonic wave with reference to the mean and maximum values of hydrostatic stress *P*_max_. The table below presents velocities of the longitudinal wave ^obs^c_p0_ determined at free state and relative mean values of hydrostatic stress *P*/*P*_max_. The measurements expressed as (*c_p_* − ^obs^*c_p_*_0_)/^obs^*c_p_*_0_ ratio of a relative increment in ultrasounds as a function of stress *P* are shown in Figure 10a. Figure 10b illustrates the relative rate of an increase of ultrasonic wave velocity rise over the relative of compressive stresses *P*/*P*_max_.

The tests showed that AAC density had an impact on velocities of ultrasonic waves, which confirmed the previous tests [14]. At stress values *P* = 0, velocities of ultrasonic waves increased in the specimens dried until constant weight. This increase was proportional to densities of AAC under stress The longitudinal wave velocity ^obs^*c_p_*_0_ in the AAC units of the minimum nominal density of 400 kg/m^3^ was equal to 1875 m/s and increased to 2225 m/s in concrete characterized by the highest density of 700 kg/m^3^. Velocities of longitudinal waves noticeably dropped as means stresses *P* increased in all the units. Under relatively low stress when 0 ≤ *P* ≤ 0.25*P*_max_, values of ultrasounds decreased by 3–9% to the value ^obs^*c_p_*_0_. At slightly higher values of hydrostatic stress 0.25*P*_max_ ≤ *P* ≤ 0.50*P*_max_ the ultrasonic wave velocities dropped by 9–16% (with reference to the base value). Higher stress values 0.50*P*_max_ ≤ *P* ≤ 0.75*P*_max_ in concrete having nominal densities of 400 and 500 kg/m^3^ caused the highest percentage drop in the velocities of ultrasonic waves by 23–24%. Ultrasonic wave velocities dropped by 13–17% in more dense masonry units made of AAC (600 and 700 kg/m^3^). In opposition to lower hydrostatic loads, no clear reduction in wave velocity was observed at the stress level preceding the failure when local cracking and crushing were found within the stress range of 0.75*P*_max_ ≤ *P* ≤ ~*P*_max_. For concretes with lower density, the velocity was reduced by 32–39%, whereas the velocity drop by 19–22% was found in concretes having density of 600 and 700 kg/m^3^. As in the tests under uniaxial stress state [11], a nearly linear drop in the relative velocity of longitudinal ultrasonic wave was observed at any density of AAC. A drop in velocity was practically 1.5–2.0 times higher than in the tests [11] on the specimens under uniaxial stress state and subjected to stress *σ*_3_. The resulting biaxial stress state confirmed the linear correlation which specified a reduced velocity of ultrasonic wave over mean hydrostatic stress. This effect was noted during the tests on AAC [11] and metals [29,32].

Table 3 presents coefficients of the linear correlation of the relative velocity of ultrasonic waves as a function of mean hydrostatic stress which are shown in Figure 10. Regression lines based on values of AE coefficients contained values of AE (*δ, η*) coefficients and density of AAC, which are illustrated in Figure 11. The coefficients were determined at moisture content of AAC *w* = 0. Additionally, values of coefficients *β*_113_, *γ*_113_ determined in the tests on uniaxial compression which are described in the paper [11], are shown in Figure 11.
(15)$$\delta =5.068\cdot 10^{-4}\rho -0.635,\;R^{2}\;=0.991,$$
(16)$$\eta =6.91\cdot 10^{-4}\rho -0.64,\;R^{2}\;=0.976$$
when $397\;\frac{\mathrm{kg}}{m^{3}}\leq \rho \leq 674\;\frac{\mathrm{kg}}{m^{3}}$.

Walls in real structures have moisture content *w* > 0 and the effect of this factor has to be taken into account. Considering the results from own research [14] and the procedure described in the paper [11], the empirical relationship was defined to determine velocities of UV waves under air-dry conditions *c_p_* (at *w* = 0) based on the equation
(17)$$\frac{c_{pw}}{c_{p}}=a{\left(\frac{w}{w_{\max}}\right)}^{2}+b\left(\frac{w}{w_{\max}}\right)+1\to c_{p}=\frac{1}{c_{pw}}\left[a{\left(\frac{w}{w_{\max}}\right)}^{2}+b\left(\frac{w}{w_{\max}}\right)+1\right],$$
where *c_pw_*—velocity of ultrasonic wave in wet AAC in the unloaded state *P* = 0; *c_p_*—velocity of ultrasonic wave in dry (*w* = 0) AAC in the unloaded state *P* = 0; *w*—relative humidity of AAC; *w*_max_—maximum relative humidity of AAC [14] calculated from the following relationship
(18)$$w_{\max}=-1.23\frac{\rho }{1000}+1.34,\;\mathrm{when}\;397\;\frac{\mathrm{kg}}{m^{3}}\leq \rho \leq 674\;\frac{\mathrm{kg}}{m^{3}}.$$
*a, b*–empirical coefficients dependent on density were
(19)$$a=9.187\cdot 10^{-4}\rho +0.932,\;\mathrm{when}\;397\;\frac{\mathrm{kg}}{m^{3}}\leq \rho \leq 674\;\frac{\mathrm{kg}}{m^{3}}.\;b=1.416\cdot 10^{-3}\rho -1.373,\;\mathrm{when}\;397\;\frac{\mathrm{kg}}{m^{3}}\leq \rho \leq 674\;\frac{\mathrm{kg}}{m^{3}}.$$

### 4.2. Stage II-Testing Models under Compression

Stage II involved small models of the masonry already used in the tests described in the previous paper [11]. The models of 500 mm × 726 mm × 180 mm in dimensions were composed of three layers of masonry units made of AAC of nominal density of 600 kg/m^3^. They were connected with thin bed joints laid in the commercial mortar with a strength *f*_m_ = 6.10 N/mm^2^ [46] and the nominal class M5 [47]. Models (nine specimens)—divided into three series marked as I, II, and III—were tested. The models differed in the presence or lack of head joints. The models of series I did not have the head joint, whereas the unfilled head joint in the central layer was at mid-length or 1/4 length of the masonry unit in other series II and III. An overall view of tests specimens of the series I, II, and III is shown in Figure 12.

All the models were subjected to monotonic compression perpendicular to the plane of bed joints by exerting the uniform increment in the shift of the testing machine piston–Figure 13. The mean normal stress *σ*_3_ was calculated as a ratio of the exerted load *F* and the area of bed face of the masonry unit *A* (*A* = 180 mm × 500 mm= 90,000 mm^2^). For two models from each series [11] velocities of ultrasonic waves *c_p_* were measured at the following values: 0, 0.25*σ*_3max_, 0.50*σ*_3max_, and 0.75*σ*_3max_. In the stage I, waves were measured using the method transmission–Figure 13a. The template was used to ensure the coaxiality of the transducers. The tests are described in details in the paper [11]. Vertical strains were measured during the tests on all the models except for I-3, II-3, III-3 series, using the digital-image correlation system ARAMIS 6M (GOM GmbH, Braunschweig, Germany) [48,49,50,51]. The main tests were preceded by determination of apparent density *ρ*_0_ (at air-dry state) and relative humidity *w* in AAC. Then the maximum moisture content *w*_max_ was calculated from the Equation (19). Table 4 presents the main results from material tests and the results from main tests as crack-inducing stress *σ*_3cr_, and maximum stress *σ*_3max_.

Considering density (*ρ*_0_ = 587–597 kg/m^3^) and relative humidity (*w* =4.5–6.0%), the research model were regarded as nearly homogeneous. In all the models a nearly proportional increase in deformations was noticed at increasing loading. Cracks were formed at failure stress of ca. > 90%. They were detected at horizontal edges of the masonry units and in the extended head joints. The failure was gentle. An increase in the width of vertical cracks and spalling of external parts of the masonry units were noticed—Figure 13b,c. The passing time *t_p_* of the ultrasonic wave was measured at defined load levels (then the strength testing machine was stopped). Calculating the velocity of the wave propagation from the relationship *c_p_* = *L*/*t_p_* (*L* = 180 mm) was the next step. The synthetic test results for all measuring points and the points located at mid-height of each masonry unit are shown in Table 5, and the partial results can be found in the paper [11]. 

As presented in the paper [11], passing time of the ultrasonic wave through the models under zero loads was characterized by some variability. The longest passing time was usually recorded in central parts of the elements. Distinct disturbances described by different passing times of the wave were noticed at vertical edges of the masonry units and at bed joints. Passing times were consistent in the central areas of the units in spite of disturbed edge areas. The obtained variation coefficient was rather low within a range of 1.4–1.6% even though all the measurements were considered (even from the disturbed areas). A clear increase in the passing time of the ultrasonic wave in all the models was observed when the loads increased up to 0.25*σ*_3max_. The coefficient of variation was rather low within a range of 1.0–1.3% as in the case of lower stress values. An increase in loads to 0.50*σ*_3max_ and 0.75*σ*_3max_ resulted in a gradual increase in the mean time of propagation for almost all measuring points. The calculated coefficients of passing time of the wave were close to the values noticed at previous loading values and amounted to ca. 1.4%.

## 5. Analysis of Test Results

### 5.1. Components of Stress State Based on the AE Effect

Values of stress *P* were determined at each measuring point using the empirical relationships which describe changes in mean hydrostatic stresses as a function of changes in the relative velocity of ultrasonic waves and propagation times of ultrasonic waves, which were determined in stage I and presented in the paper [7]. Mean values of hydrostatic stress *P* expressed as the maps of stress at different stress levels (0.25*σ*_3max_, 0.50*σ*_3max_, 0.75*σ*_3max_) are shown in Figure 14, Figure 15 and Figure 16.

The distribution of mean hydrostatic stresses *P* in all the test models indicated the predominating compression (*P* > 0) in the masonry units. Mean hydrostatic stresses were clearly decreasing in some areas adjacent to the head joints. Only at some individual points did mean hydrostatic stress represent tension (*P* < 0). 

By reference to the paper [11], the qualitative analysis for the obtained results was performed in a comprehensive way using all the test results and then was constrained to a limited number of points. The comprehensive method included *n* = 315 (the model of series I) or 308 (the models of series II or III) measured passing times of ultrasonic wave at each analyzed stress level. The results for the clearly disturbed areas were also taken into account. In the method using a limited number of points stress was estimated only on the basis of the points located in the central area of the masonry units. In that way, the measuring points were considerably reduced to 45 for the model I, and to 44 for the models of series II and III.

At first, the velocity of ultrasonic waves was determined under air-dry conditions according to the following relationship (17). A relative difference in the passing time of the ultrasonic wave was then determined at other stress values. Determination of the acoustoelastic coefficient *δ_P_* from the Equation (15) was the next step. At the end the stress *P* was obtained from the converted relationship (10). Table 6 demonstrates the calculated stresses.

Values of coefficients *δ_P_* depended on the density of AAC, however, these differences were relatively small (*δ_P_* = −0.0635–−0.0640 mm^2^/N). Mean values of hydrostatic stress were evidently increasing with an increase in vertical stress values, which showed that compressive stress predominated in the compressed wall. The stress values calculated for the individual models were close to each other only when stress values were relatively low, that is, 0.25*σ*_3max_ and 0.50*σ*_3max_. Stresses in the model III-1 determined by the AE differed by maximum 12%. At 0.75*σ*_3max_ the stress values did not differ by more than 8%.

The same procedure was repeated in the method based on the limited number of results (from central areas of the masonry units). Analogous to the method, which was based on all the test results, velocities of ultrasonic waves under air-dry conditions were determined at first, and then a relative difference in the passing time of the ultrasonic wave and the stress values *P* were calculated from the converted relationship (10). The coefficient *δ_P_* was the same as the value specified in Table 6. The obtained values of hydrostatic stress *P* are presented in Table 7.

The stress values were much lower at a limited number of measuring points. When stresses were the lowest, that is, equal to 0.25*σ*_3max_, the stresses determined with the AE method at the minimum number of points were lower by no more than 31% (the model II-1). Mean hydrostatic stresses at higher stresses (0.50*σ*_3max_ and 0.75*σ*_3max_) were underestimated by a maximum of 18%.

By knowing mean hydrostatic stresses and stresses *σ*_3_ determined from the Equation (12) and presented in the paper [11], horizontal stresses *σ*_1_ could be determined from the relationship
(20)$$P=\frac{1}{3}\left(\sigma_{1}+\sigma_{2}+\sigma_{3}\right)=\frac{1}{3}\left(\sigma_{1}+\sigma_{3}\right)\to \sigma_{1}=3P-\sigma_{3},$$
where *P*—mean hydrostatic stress, *σ*_3_—normal stress perpendicular to the plane of bed joints, *σ*_1_—normal stress parallel to the plane of bed joints.

The values of stress *P* and stress *σ*_3_ shown in Table 6 and Table 7 and presented in the paper [11], were the base to determine stresses *σ*_1_ which are summarized in Table 8.

### 5.2. Numerical FEM Model

The numerical FEM model was necessary to perform the comprehensive analysis of the determined mean values of hydrostatic stress *P* and normal stress *σ*_1_ (determined indirectly on the basis of known values of stress *σ*_1_). This model was used to determine mean values of hydrostatic stress from the components of the stress state. As it was demonstrated in the paper [11], the defined relationships between stress and strain were similar to the linear relationship. Hence, the linear-elastic FEM micro-model was sufficient for that purpose. The model included nominal geometric dimensions and boundary conditions. The model was 726 mm high, 500 mm wide, and 180 mm thick. It was supported along its bottom edge using the roller supports in each node, except for the middle one with the blocked horizontal movement. Five-node finite elements with 4 degrees of freedom for each node were used for calculations in a plane stress state (2D, PSS). The masonry units were modelled separately, for which the modulus of elasticity was *E*_B_ = 2039 N/mm^2^ and Poisson’s ratio was ν_B_ = 0.21 (cf Table 1). Mortar in joints was also modelled separately, and the finite elements took the following parameters *E*_m_ = 6351 N/mm^2^ and ν_m_ = 0.18 [52]. Due to linear elasticity of the FEM models, the model was subjected to unit loads *q* = 1 kN/m, and the stress values at higher loads were determined using superpositioning of load states. The numerical FEM models are shown in Figure 17. The calculated data presented as the maps of vertical stresses *σ*_3_ and *σ*_1_ of unit loads are illustrated in Figure 18. 

The stress–strain relationships for all the test elements and the FEM models were compared as shown in Figure 19. These curves indicate that the behavior of the test models was almost linear until the moment of cracking. Strains began to increase much faster than in the linear-elastic FEM model when the stresses were >0.75*σ*_3max_. Differences in calculated and experimentally determined moduli of elasticity did not exceed 10%.

The data for components of the stress states *P*, *σ*_1_, *σ*_3_ obtained on the basis of the FEM calculations are compared in Table 9. The results are compared in Table 10.

The maximum difference in mean stresses *σ*_3_ perpendicular to the plane of bed joints, which were determined for all the measuring points, was 3% at the stress levels 0.25*σ*_3max_–0.50*σ*_3max_. The biggest difference was observed under compressive stress equal to 0.75*σ*_3max_. When the number of measurements was limited to central areas of the masonry units, significantly greater differences were noticed, The highest mean overestimation of the results exceeding 32% was found at the stress level of 0.75*σ*_3max_. Almost the same results were obtained for the hydrostatic stress *P* in the wall. For all the measuring points, the differences did not exceed 3% at the stress levels of 0.25*σ*_3max_–0.50*σ*_3max_. Mean stresses were greater by 31% also under higher compressive stress equal to 0.75*σ*_3max_.

The greatest variation of the results was found under the stresses *σ*_1_ which were parallel to the plane of bed joints. A higher number of measurements caused in this case a higher degree of inconsistency between the results. Under the compressive stresses 0.25*σ*_3max_–0.50*σ*_3max_, the stress values were overestimated by ca. 44–50%. An increase in mean values of stresses to the level of 0.75*σ*_3max_ caused that the overestimation of stresses was reduced to ca. 19%. The best results were obtained for the limited number of measuring points. Then, at the stress levels of 0.25*σ*_3max_ and 0.75*σ*_3max_, the stresses determined with the NDR technique did not significantly vary from the stresses obtained from the FEM calculations. The highest overestimation of the stresses of the order of 28% was found for mean stresses equal to 0.50*σ*_3max_.

The best agreement with the FEM calculations was reached when the maximum number points were used for vertical stresses *σ*_3_ and mean hydrostatic stresses *P*. The measurements limited to central areas of the masonry units resulted in bigger differences in the results when compared to the numerical results. The contradictory tendency was noticed for the stresses *σ*_1_, under which the biggest differences in the results were obtained when the maximum number of points were used. Limiting the measurements only to the central areas caused a clear drop in the stress values which were empirically determined.

The results were obtained from the methodology of determining the coefficients AE (*β*_113_ *η*_P_), which was conducted on relatively small specimens subjected to the load which eliminated additional stress components and boundary disorders. The stress distribution in real masonry structures (on which the main tests were performed) is significantly disturbed by the presence of head and bed joints, the shape, and interaction with other masonry units. The results were close to the FEM calculations when the measurements were taken in the central area of the masonry units at the least disturbed stress state. A narrower spread of the calculated and test results was the immediate effect. It should be remembered that the plane stress state assumed for the analyses is observed locally in central areas of the masonry units. Additional stresses *σ*_2_ ≠ 0 perpendicular to the front plane of the masonry are found in the edge and support areas, which has an impact on mean hydrostatic stresses. The proposed procedure cannot be applied for the whole range of stress values without its prior calibration. At relatively low stresses 0.25*σ*_3max_ and 0.50*σ*_3max_, the test results were similar to the calculated results. The most significant differences were obtained for the stresses of 0.75*σ*_3max_, and at this level NDT tests can be performed.

### 5.3. Model Update

It is more favorable to perform in practice only the tests restrained to central areas of the masonry units. Such an approach reduces the effect of disturbances created at the element edges due to the presence of bed and head joints. As shown in point 6.2, the NDT technique based on the AE effect and the stresses determined for the central areas of masonry units are expected to provide inconsistency in all the determined stresses. Assuming that the results obtained from the FEM calculations correctly estimate the stress values in the masonry units, the stress values *σ*_3_ in the masonry units were at first corrected. For that purpose, mean quotients presented in Table 9 were applied. It was the base to calculate the mean quotient of stresses determined by the NDT and FEM techniques. On this basis, the mean coefficient equal to *α*_3_ = 0.75 was determined. Coefficients of stresses *P* and *σ*_1_ were determined similarly. These values were *α_P_* = 0.76 and *α*_1_ = 1.09. The update empirical values to determine stress in the wall can be expressed as
(21)$$\sigma_{3}=\frac{1}{\alpha_{3}}\frac{\left(c_{p}-c_{p0}\right)}{c_{p0}\beta_{113}},$$
(22)$$P=\frac{1}{\alpha_{P}}\frac{\left(c_{p}-c_{p0}\right)}{c_{p0}\delta_{P}},$$
(23)$$\sigma_{1}=\frac{1}{\alpha_{1}}\left(3P-\sigma_{3}\right).$$

The results obtained by the NDT technique before and after validation and by the FEM methods are compared in Figure 20, Figure 21 and Figure 22.

The stresses *σ*_3_ and *P* in the updated model were underestimated by no more than 15%. On the other hand, the underestimation of the stresses *P* parallel to the plane of head joints *σ*_1_ did not exceed 6%.

The described validation resulted in mean stresses in the wall which were comparable to the data determined by the FEM technique after taking at least *n* > 44 measurements in the central parts of the wall. However, taking the measurements at so many points (at a relatively low variation) can be troublesome in practice. That is why it is necessary to specify the minimum number of measuring points, at which the obtained results are reliable with reference to the defined confidence level [53]. Therefore, the minimum number of measuring points was define assuming that:the general population had the normal distribution N(*μ*, *σ*),the variance *σ* of the general population was unknown at the known standard deviation for the small sample, which was taken as $s=\nu_{\mathrm{test}}\bar{x}$ (*ν*_test_ = 15%—the coefficient of variation corresponding to the results from the in-situ tests),*α* = 0.05—the confidence level,the relative error was defined at the level 0.5*α* = 0.0025. The absolute value was taken as *d* = 0.5α $\bar{x}$,the minimum number of samples [53] were determined from the relationship $n_{0}={\left(t_{\alpha ,n-1}s/d\right)}^{2}$, where *t_α_*_,*n*−1_ = 2.017—the parameter of a two-tailed T distribution at *n* − 1 degrees of freedom, *n* = 44—the number of samples to determine the number of samples.

Based on these assumptions, the calculations were performed under the stresses equal to 0.75*σ*_3max_. Only the stresses *σ*_3_ and *P* calculated from the relationships (21) and (23) were considered. It was not necessary to specify the number of samples on the basis of the stress *σ*_1_ as it was not an independent variable. The obtained number of samples is shown in Table 11. 

When the tests were focused on determining compressive stress *σ*_3_ during the in-situ tests on the wall made of AAC masonry units, the minimum number of measurements was estimated to be *n*_0_ = 8. On the other hand, when the aim of the tests is to determine the complex state of stress, the minimum number of measurements should not be lower than *n*_0_ = 23. When the results were expressed in 1 m^2^ of the wall, then the minimum number of measuring points required to determine stresses *σ*_3_ should not be lower than *n*_0_ = 8·(1/0.726·0.5) = 22 measurements/m^2^, and in case of mean hydrostatic stresses *n*_0_ = 23·(1/0.726·0.5) = 61 measurements/m^2^.

The proposed update method was intended to determine mean stresses in the wall, which were crucial for diagnostic tests for structures. Development of the complete model which can be used to define characteristics and design values to verify the estimated structural safety, should include the non-linear FE model and the application of FORM procedures [54,55].

## 6. Conclusions

This paper is a continuation of the tests [11,14] concerning the use of the ultrasonic (UPV) techniques, in particular the acoustoelastic (AE) method to detect stresses in a structure by means of the non-destructive technique (NDT). The tests were focused on the commercially produced autoclave aerated concrete (AAC) which is characterized by high homogeneity and repeatability of the parameters. Considering different purposes, the tests were carried out in two stages. In Stage I, the test procedure was specified and the acoustoelastic coefficient *δ_P_* was determined. This coefficient specified the relationships between the mean hydrostatic stresses *P* and the velocity of the longitudinal ultrasonic wave propagation *c_p_*.

The non-standard cuboidal specimens 180 × 180 × 120 mm were used for the calibration purposes. They were tested at the in-house developed test stand [37] which can be used to exert the biaxial compression. Based on the tests on AAC of different densities, the impact of relative humidity *w* and density ρ was included using the correlations presented in paper [14]. These considerations resulted in formulating the relationship δ_P_(ρ). Verification of the discussed procedure was performed in Stage II, in which the complex stress state was to be determined. This stage based on the results from previous test [11] performed on small AAC walls having a nominal density of 600 kg/m^3^. The models differed in the position of head joints without mortar and were classified into series I, II, and III. The measured velocities of ultrasonic wave propagation were analyzed under various compressive stresses: 0.25*σ*_3max_, 0.50*σ*_3max_ and 0.75*σ*_3max_. The performed measurements (*n* = 308–315) were used to define the coefficients AE δ_P_ = −0.1632–−0.0281. The data obtained from the AE method were compared with the data calculated for the linear-elastic FEM models of the walls. For the mean values of hydrostatic stress *P,* the stresses were underestimated at the order of 3% at 0.25*σ*_3max_. Under higher compressive stresses 0.50*σ*_3max_, the stresses P obtained by the AE method were greater by 2% than the calculated mean values. Under the highest analyzed stresses equal to 0.75*σ*_3max_, the empirically determined stresses were greater by over 19% than the calculated values. By knowing the stresses *P* and the stresses *σ*_3_ perpendicular to the plane of head joins presented in the paper [11], the stresses *σ*_1_ could be determined. These results were compared with the values obtained by the FEM calculations under various compressive stresses. Each time the values were overestimated. The stress values *σ*_1_ at 0.25*σ*_3max_ were overestimated by 44%. An increase in vertical loads to the values of 0.50*σ*_3max_ and 0.75*σ*_3max_ caused that the stress values determined with the AE method were greater by 50% and 19% compared to the data obtained from the FEM method. These discrepancies were caused by disorders of the stress state in the real structure and they considerably differed from the stress state, under which the coefficient AE (*β*_113_ and δ_P_) was determined. 

It is not effective to use so many measuring points in practice (as a high number of points and results from the measurements have to be prepared and captured). For that reason, it was suggested that the measuring points were constrained only to the central areas of each masonry units which reduced the number of measurements to *n* = 45 and 44. 

A similar comparison as for all the measurements produced considerably higher underestimations of the mean stresses *σ*_3_ by 13–32%, and the stresses *P* by 3–19%. These values are not desirable taking into account safety of the structure. Hence, a decision was made to validate the model using the numerical FEM model by defining the coefficients *α*_3_ = 0.75, *α*_P_ = 0.76, and *α*_1_ = 1.09. The stresses in the validated model were underestimated by no more than 15% under the stresses *σ*_3_ and *P*. On the other hand, under the stresses *P* parallel to the plane of head joints *σ*_1_ the underestimation did not exceed 6%. 

In summary:the acoustoelastic (AE) method was confirmed to be applied to mean hydrostatic stresses in AAC,the relationships between the acoustoelastic coefficient δ_P_ and AAC density and moisture content AAC were established,the performed measurements of the velocity of ultrasonic wave propagation were used to quite precisely determine the mean hydrostatic stresses in the wall (when compared to the FEM calculations) when the number of measuring points was high,a reduction in the measuring points significantly underestimated the mean hydrostatic stresses,the method validation considerably diminished differences between the experimentally obtained results and the calculations. The maximum overestimation of stress values did not exceed 15%, and the underestimation was at the level of 6%.an empirical nature of the employed method constraints possible applications to the complete range of standard stresses in the masonry. The reliable estimation of the mean stresses for the model validated can be used even to the level of <0.75*σ*_3max_.

Also, the minimum number of measurements were defined to ensure reliability of the results at the pre-determined measurement error at the specified level of confidence. If the tests are to measure normal stresses in the plane of bed joints, then the minimum required number of measuring points is 22 measuring points/m^2^. The tests focused on the analysis of the complex state of stresses require the minimum number of measurements equal to 61 measurements/m^2^. 

Specifying the detailed guidelines for in-situ tests for structures at the present stage of analyses of masonry structures is impossible. It is required to conduct additional tests on slender walls to determine the bending effect (varied stress state in the wall) and to improve the methodology of selecting the measuring points. The selection method of measuring points used to evaluate both the complex and the uniaxial stress state [11] may prove to be inadequate for bending. The double-sided access to the structure can be another problem. Hence, further tests are planned to be performed on the AE coefficient AE (β_133_) in the AAC wall with one-sided access.

## Figures and Tables

**Figure 1 materials-14-03459-f001:**
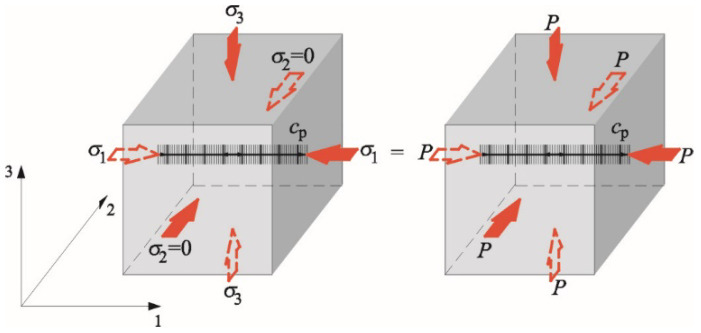
Directions of the stress and the ultrasonic wave in the isotropic material at the state of hydrostatic compression.

**Figure 2 materials-14-03459-f002:**
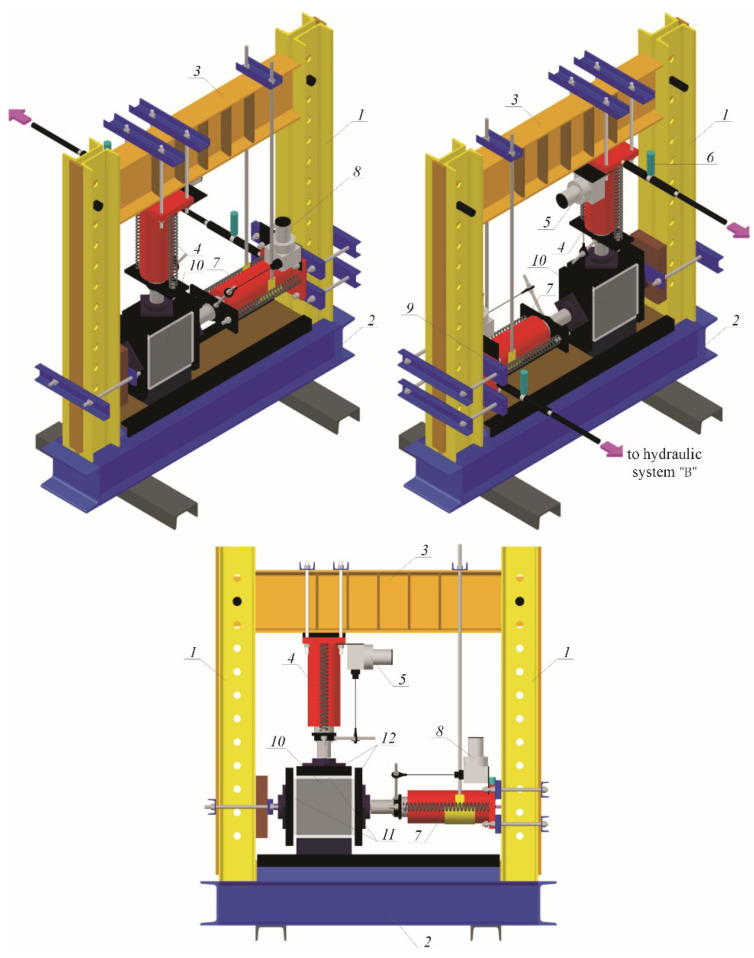
Test stand for measuring the acoustoelastic effect under the biaxial stress state (described in the text).

**Figure 3 materials-14-03459-f003:**
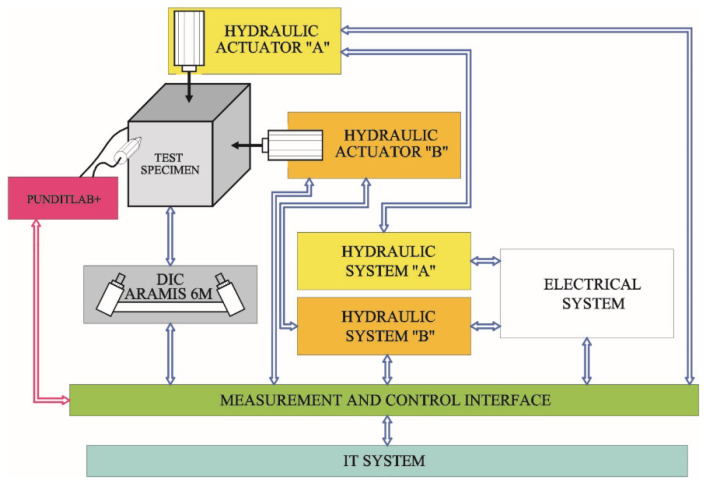
Block diagram of the test stand.

**Figure 4 materials-14-03459-f004:**
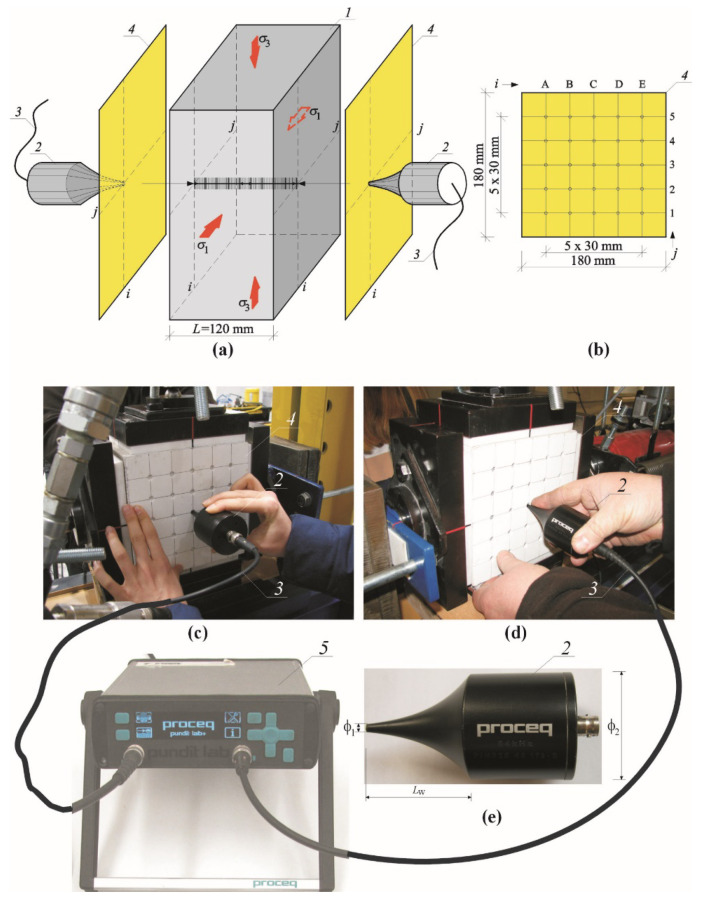
Measurements of ultrasonic wave velocity in biaxially compressed specimens: (**a**) components of stress states and the position of the measuring template; (**b**) geometry of the measuring template; (**c**,**d**) the test specimen; (**e**) the exponential transducer; 1—the AAC specimen 180 mm × 180 mm × 120 mm, 2—exponential transducers, 3—cables connecting transducers with recording equipment, 4—the measuring template, 5—PUNDIT LAB+ recording equipment.

**Figure 5 materials-14-03459-f005:**
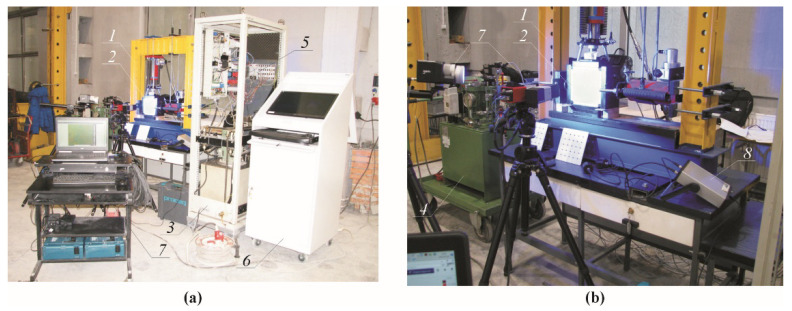
View of the test stand in operation: (**a**) overall view, (**b**) specimen view; 1—the test stand with the fixed actuators; 2—a test element; 3—the hydraulic system ‘A’; 4—the hydraulic system ‘B’; 5—the measurement and control interface; 6—IT system; 7—cameras of the ARAMIS 6M system; 8—PUNDIT LAB+ instrument.

**Figure 6 materials-14-03459-f006:**
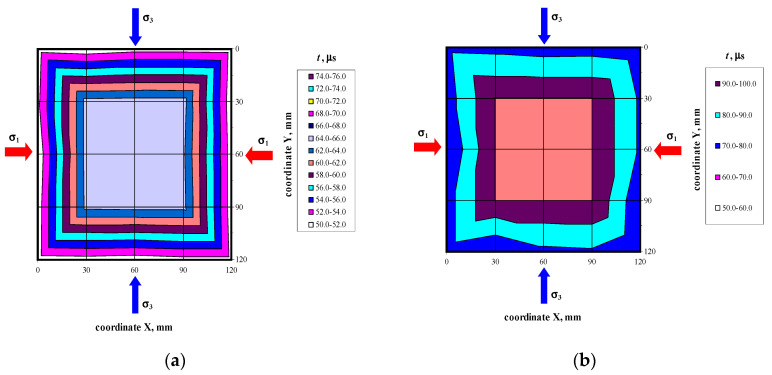
Maps of passing time the ultrasonic wave in the model 400/1 at selected loading levels: (**a**) *P* = 0, (**b**) *P* = *P*_max_.

**Figure 7 materials-14-03459-f007:**
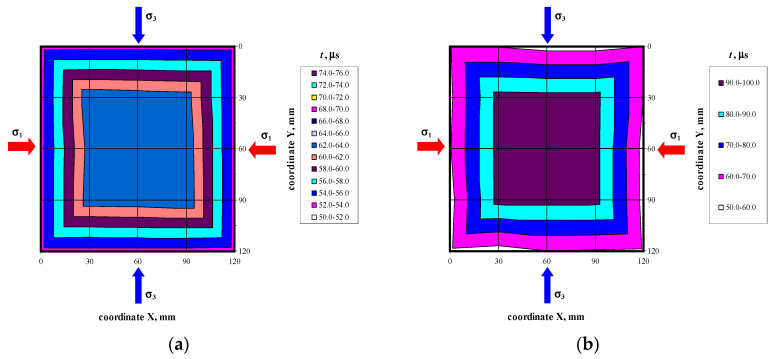
Maps of passing time the ultrasonic wave in the model 500/1 at selected loading levels: (**a**) *P* = 0, (**b**) *P* = *P*_max_.

**Figure 8 materials-14-03459-f008:**
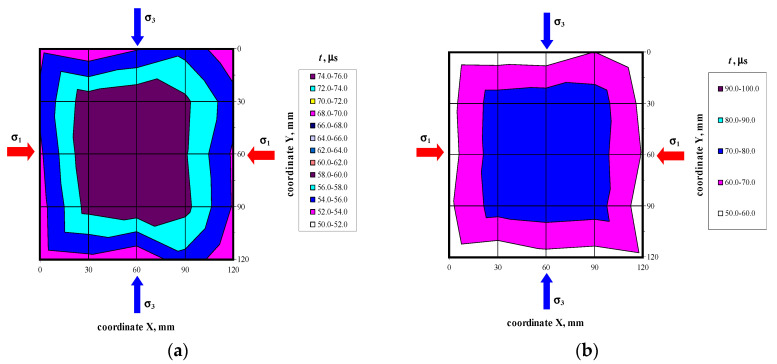
Maps of passing time the ultrasonic wave in the model 600/1 at selected loading levels: (**a**) *P* = 0, (**b**) *P* = *P*_max_.

**Figure 9 materials-14-03459-f009:**
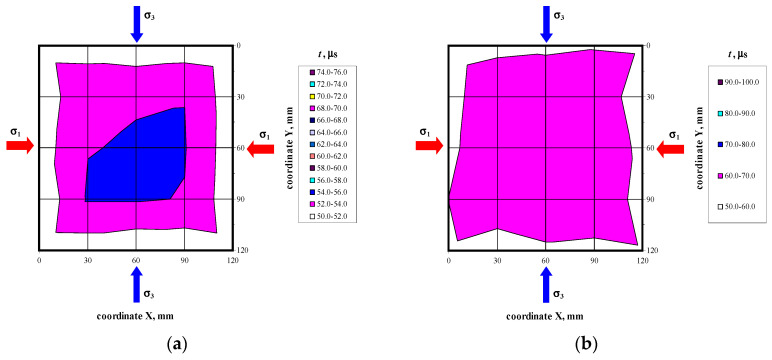
Maps of passing time the ultrasonic wave in the model 700/1 at selected loading levels: (**a**) *P* = 0, (**b**) *P* = *P*_max_.

**Figure 10 materials-14-03459-f010:**
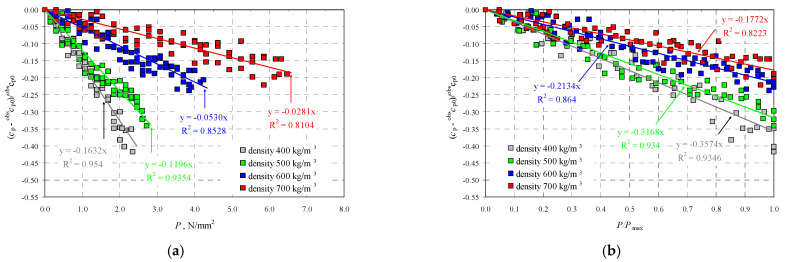
Results from measuring velocity of the longitudinal ultrasonic wave: (**a**) relative change in velocity of longitudinal wave as a function of compressive stress, (**b**) relative change in velocity of longitudinal wave as a function of relative compressive stresses.

**Figure 11 materials-14-03459-f011:**
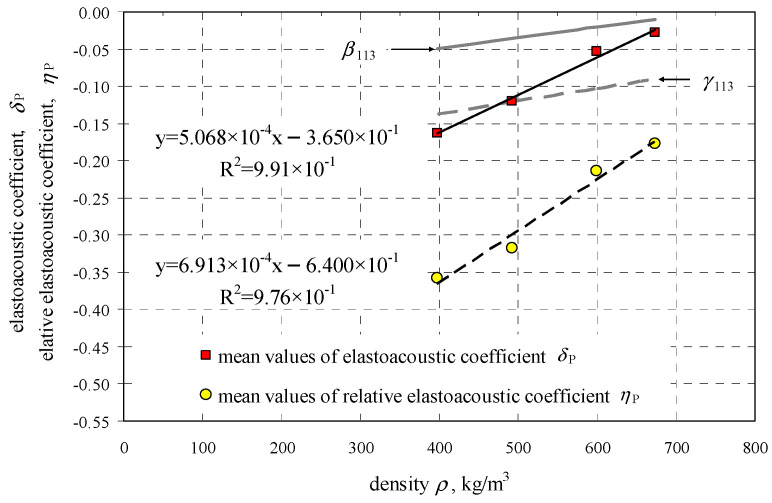
Values of coefficients δ and η as a function of AAC density.

**Figure 12 materials-14-03459-f012:**
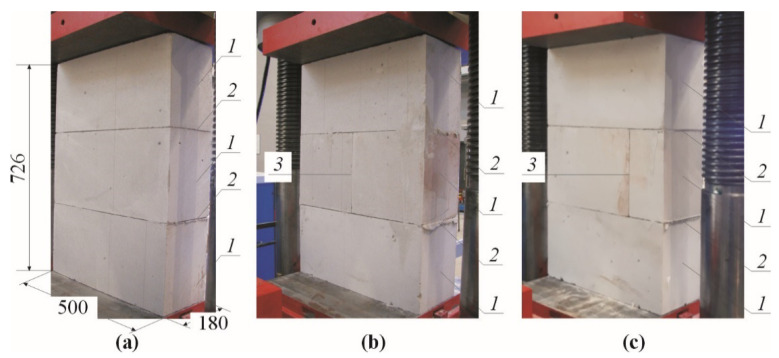
Geometry of models made of AAC tested in stage II: (**a**) models of series I without head joint, (**b**) models of series II with head joint at mid-length of the masonry unit, (**c**) models of series III with head joint at 1/4 length of the masonry unit; 1—masonry units, 2—bed joints, 3—head joints.

**Figure 13 materials-14-03459-f013:**
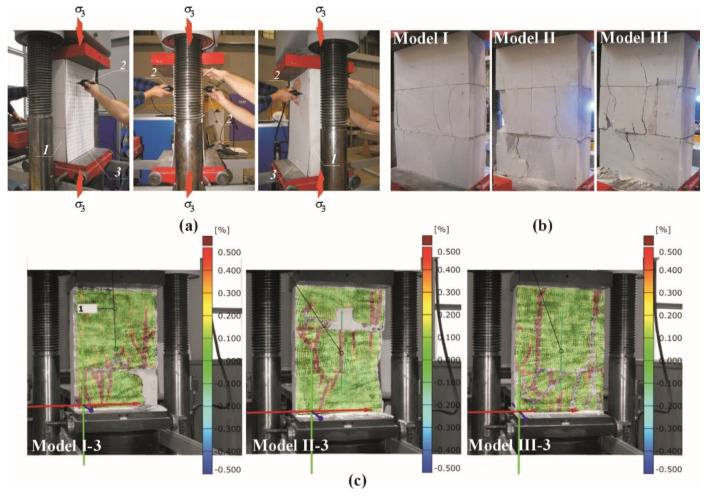
The procedure employed in Stage II to test the AAC wall models: (**a**) measurements of velocity of the ultrasonic wave at different stress value *σ*_3_, (**b**) selected models at failure, (**c**) vertical strains of selected wall models under stress *σ*_3max_; 1—masonry units, 2—ultrasonic transducers, 3—templates to arrange symmetrically ultrasonic transducers.

**Figure 14 materials-14-03459-f014:**
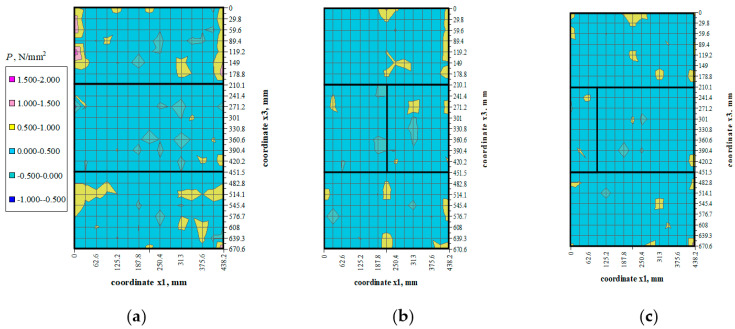
Mean hydrostatic stress values *P* at load *σ*_3_ = 0.25*σ*_3max_: (**a**) model I-1, (**b**) model II-1, (**c**) model III-1.

**Figure 15 materials-14-03459-f015:**
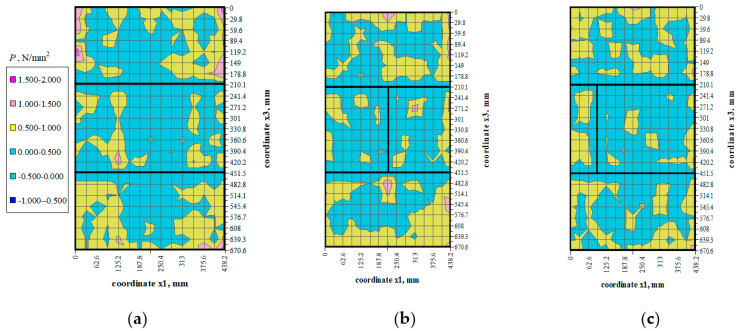
Mean hydrostatic stress values *P* at load *σ*_3_ = 0.50*σ*_3max_: (**a**) model I-1, (**b**) model II-1, (**c**) model III-1.

**Figure 16 materials-14-03459-f016:**
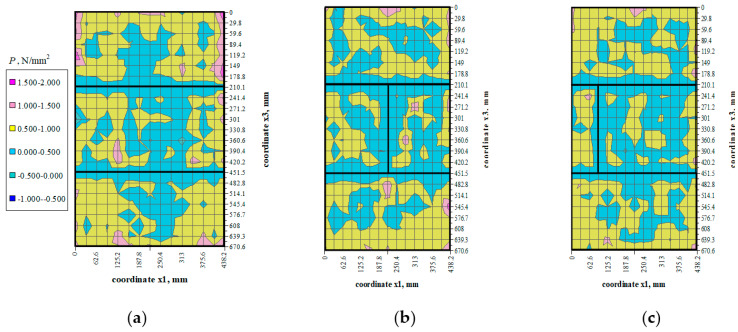
Mean hydrostatic stress values *P* at load *σ*_3_ = 0.75*σ*_3max_: (**a**) model I-1, (**b**) model II-1, (**c**) model III-1.

**Figure 17 materials-14-03459-f017:**
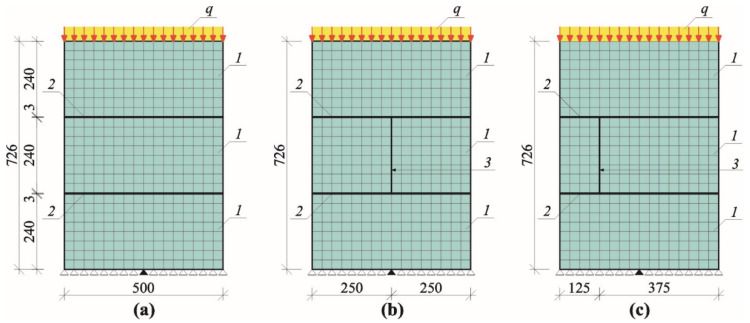
FEM models for the test walls: (**a**) models of series I without head joint, (**b**) models of series II with head joint at mid-length of the element, (**c**) models of series III with head joint at 1/4 length the masonry unit *1*—masonry unit; *2*—mortar; *3*—unfilled head joint.

**Figure 18 materials-14-03459-f018:**
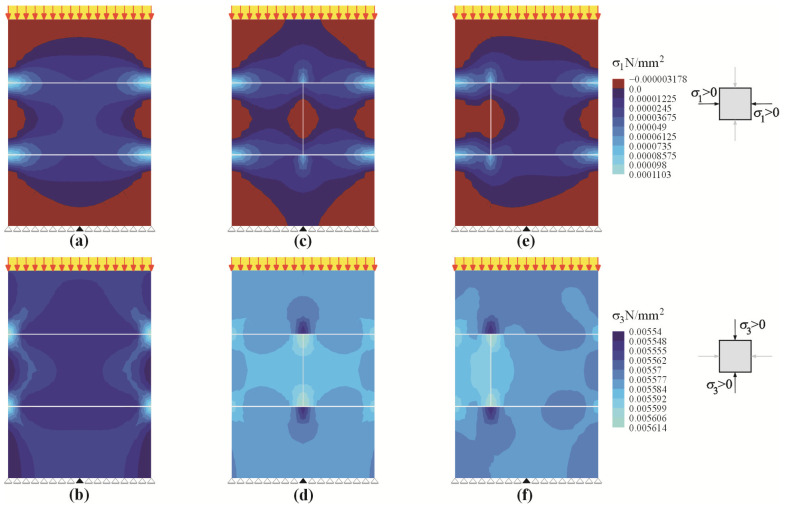
FEM calculations for the test walls: (**a**) stresses *σ*_x_ in the model of series I without a head joint, (**b**) stresses *σ*_y_ in the model of series I without a head joint, (**c**) stresses *σ*_x_ in the model of series II with the head joint at mid-length of the element, (**d**) stresses *σ*_y_ in the model of series II with the head joint at mid-length of the element, (**e**) stresses *σ*_x_ in the model of series III with the head joint in 1/4 length of the masonry unit, (**f**) stresses *σ*_y_ of the model of series III with the head joint in 1/4 length of the masonry unit.

**Figure 19 materials-14-03459-f019:**
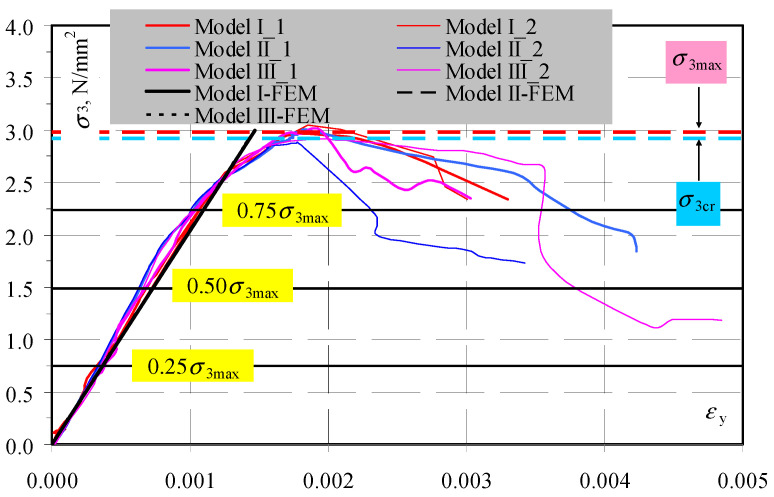
Compared relationships between stress and strain (*σ*_3_–ε_y_) for all tested models and numerical FEM models.

**Figure 20 materials-14-03459-f020:**
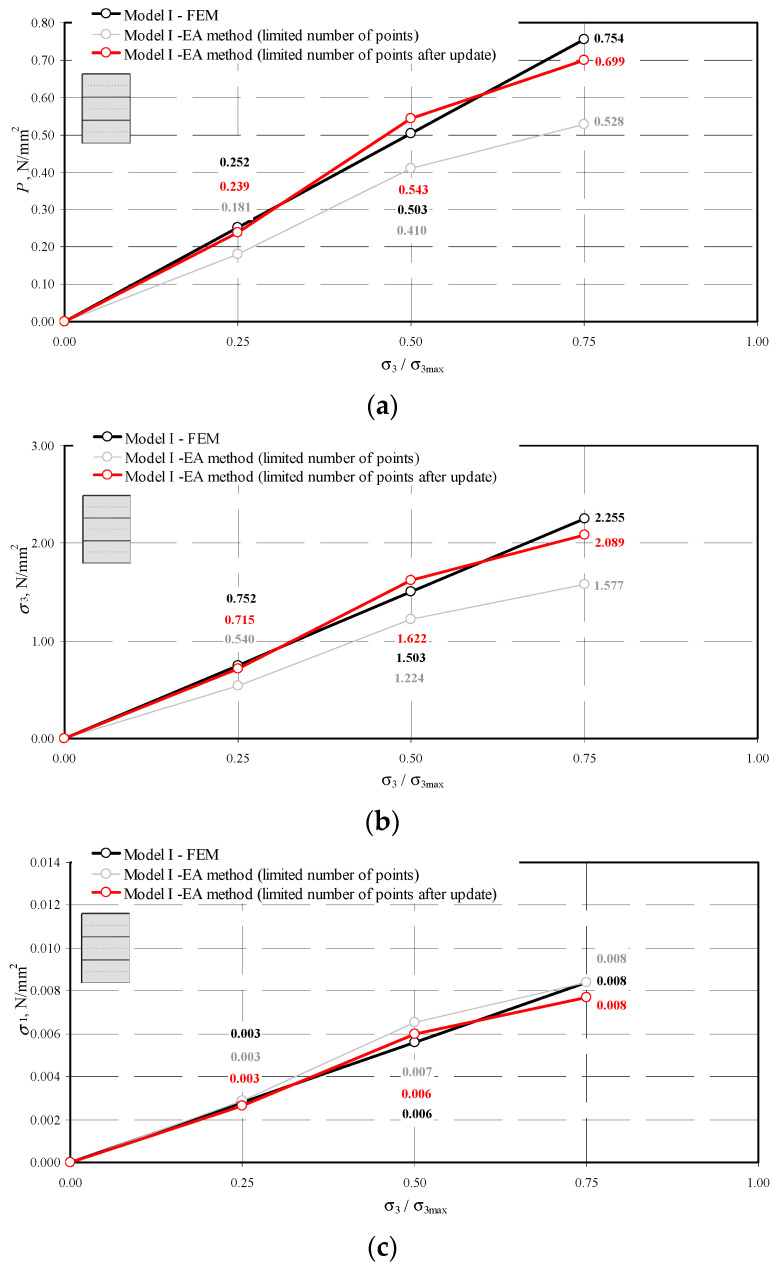
Comparison of stress values determined by the NDT and FEM methods for the model I-1: (**a**) mean hydrostatic stress; (**b**) normal stress perpendicular to the plane of bed joints; (**c**) normal stress parallel to the plane of bed joints.

**Figure 21 materials-14-03459-f021:**
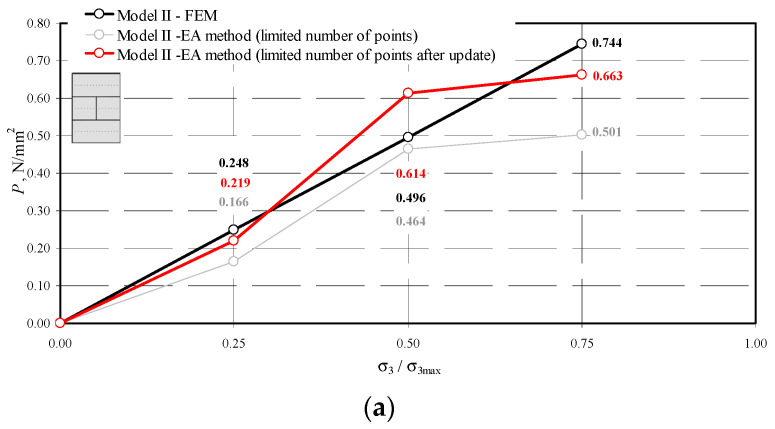
Comparison of stress values determined by the NDT and FEM methods for the model II-1: (**a**) mean hydrostatic stress; (**b**) normal stress perpendicular to the plane of bed joints; (**c**) normal stress parallel to the plane of bed joints.

**Figure 22 materials-14-03459-f022:**
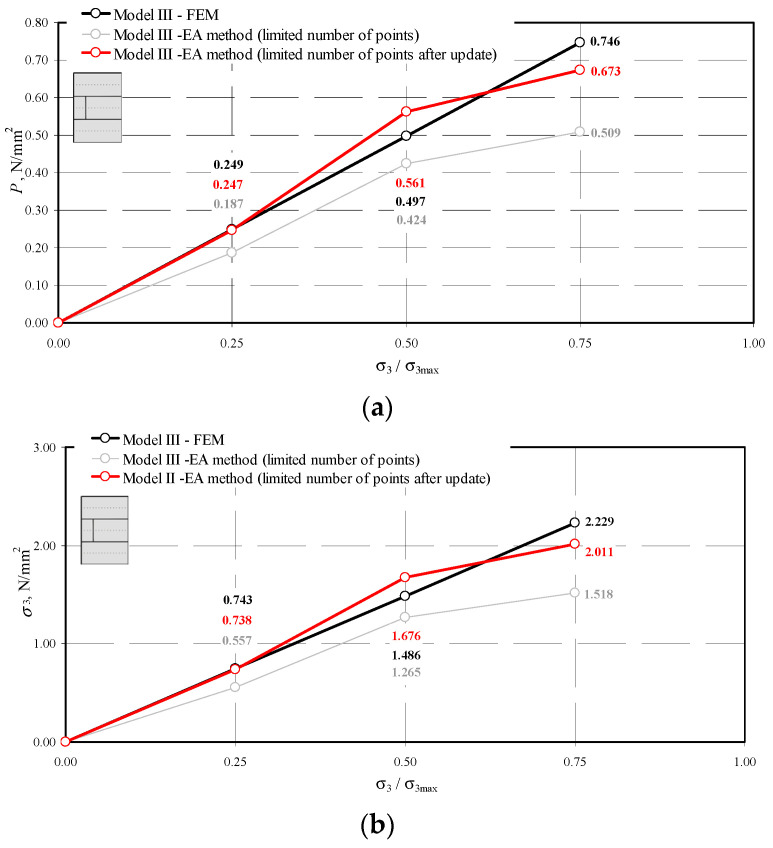
Comparison of stress values determined by the NDT and FEM methods for the model III-1: (**a**) mean hydrostatic stress; (**b**) normal stress perpendicular to the plane of bed joints; (**c**) normal stress parallel to the plane of bed joints.

**Table 1 materials-14-03459-t001:** Fundamental characteristics of masonry units as defined in the papers [11,14].

No.	Nominal Class of Density kg/m^3^ Acc. to [11]	Density Range of AAC, kg/m^3^ Acc. to [11]	Mean Density *ρ*_0_, kg/m^3^ (C.O.V) Acc. to [14]	Mean Modulus of Elasticity *E*, N/mm^2^ (C.O.V) Acc. to [11]	Mean Poisson’s Ratio *ν*, (C.O.V) Acc. to [11]	Compressive Strength of AAC *f*_B_, N/mm^2^ (C.O.V) Acc. to [14]
1	400	375–446	397 (6%)	1516 (9.6%)	0.19 (7.9%)	2.88
2	500	462–532	492 (3%)	2039 (8.9%)	0.21 (8.7%)	3.59
3	600	562–619	599 (2%)	2886 (10.5%)	0.20 (8.5%)	5.05
4	700	655–725	674 (3%)	4778 (10.1%)	0.19 (9.2%)	8.11

**Table 2 materials-14-03459-t002:** Test results for ultrasonic wave velocity in AAC under various mean hydrostatic stresses determined at central points (B1–B3, C1–C3, D1–D3, E1–E3) of each specimen.

No.	Mean Density *ρ*, (Nominal Class of Density) kg/m^3^	Mean Compressive Stress *P*, N/mm^2^	Mean Relative Compressive Stress *P/P*_max_	Mean Path Length *L*, mm	Mean Passing Time of Wave *t*, µs	Mean Ultrasonic Velocity *c_p_* = *L/t*, m/s	$\frac{\left(c_{p}-c_{p0}\right)}{c_{p0}}$	COV, %
1	2	3	4	5	6	7	8	9
1	397 (400)	0	0	120.1	64.7	^obs^*c_p_*_0_ = 1875	0	1.7%
2		0.51	0.23		70.6	1704	−0.09	2.1%
3		1.13	0.52		76.5	1572	−0.16	1.4%
4		1.65	0.75		87.0	1387	−0.26	0.5%
5		2.19	1		105.2	1145	−0.39	2.3%
6	492 (500)	0	0	119.9	63.4	^obs^*c_p_*_0_ = 1893	0.00	2.1%
7		0.62	0.23		69.3	1732	−0.08	1.9%
8		1.34	0.51		78.3	1534	−0.19	1.6%
9		2.01	0.76		82.8	1451	−0.23	1.1%
10		2.65	1		93.4	1286	−0.32	1.7%
11	599 (600)	0	0	120.1	59.1	^obs^*c_p_*_0_ = 2031	0.00	1.9%
12		0.98	0.24		61.3	1960	−0.03	3.1%
13		2.01	0.50		66.7	1800	−0.11	2.7%
14		3.03	0.76		70.8	1695	−0.16	2.2%
15		4.01	1		75.6	1588	−0.22	2.4%
11	674 (700)	0	0	120.2	54.0	^obs^*c_p_*_0_ = 2225	0.00	2.1%
12		1.54	0.25		57.7	2083	−0.06	1.4%
13		3.19	0.51		59.1	2032	−0.09	1.8%
14		4.73	0.75		62.0	1936	−0.13	1.9%
15		6.30	1		66.8	1799	−0.19	3.1%

**Table 3 materials-14-03459-t003:** Values of AE coefficients for concrete of specific densities.

No.	Mean Density *ρ*, (Nominal Class of Density) kg/m^3^	AE Coefficient *δ_P_*, m^3^/kg	Relative Coefficient *η_P_*
1	2	3	4
1	397 *(400)*	−0.1632	−0.3574
2	492 *(500)*	−0.1196	−0.3168
3	599 *(600)*	−0.0530	−0.2134
4	674 *(700)*	−0.0281	−0.1772

**Table 4 materials-14-03459-t004:** Summary of mean results from the tests on the models.

Series	Mean Density *ρ*_0_, kg/m^3^	Moisture Content *w*, %	Maximum Moisture Content (17) *w*_max_, %	Mean Compressive Stress Inducing Cracks *σ*_3cr_, N/mm^2^ (COV)	Maximum Mean Compressive Stress *σ*_3max_, N/mm^2^ (COV)
1	2	3	4	5	6
I	592	5.20%	61.2%	2.89	3.01
	(0.43%)	(14.5%)	(0.57%)	(1.1%)	(1.3%)
II	595	5.63%	61.1%	2.95	2.96
	(0.34%)	(11.3%)	(0.90%)	(2.8%)	(2.6%)
III	590	5.33%	61.4%	2.90	2.97
	(0.59%)	(3.90%)	(0.73%)	(3.3%)	(1.9%)

**Table 5 materials-14-03459-t005:** Results from measuring propagation of ultrasonic waves.

Series	No. of Measuring Points in Each Loading Step, *n*	Time of Ultrasonic Wave Passing at Different Levels of Loading, *t*_pmv_, μs (COV)			
		0	0.25*σ*_3max_	0.50*σ*_3max_	0.75*σ*_3max_
1	2	3	4	5	6
I-1	315	90.8	92.2	93.9	94.4
		(1.4%)	(1.3%)	(1.4%)	(1.4%)
	45	91.2	92.3	93.6	94.3
		(1.3%)	(1.1%)	(1.3%)	(1.2%)
II-1	308	89.2	90.6	92.2	92.5
		(1.6%)	(1.2%)	(1.1%)	(1.1%)
	44	89.6	90.5	92.2	92.4
		(1.5%)	(1.2%)	(0.9%)	(0.9%)
III-1	308	88.8	90.2	91.6	92.1
		(1.4%)	(1.2%)	(0.9%)	(0.9%)
	44	89.1	90.2	91.5	92.0
		(1.2%)	(0.8%)	(0.7%)	(0.8%)

**Table 6 materials-14-03459-t006:** Calculated mean values of hydrostatic stress in the wall using all measuring points.

Model	Number of Measurements *n*	0.25*σ*_3max_			0.50*σ*_3max_			0.75*σ*_3max_		
		$\frac{\left(c_{p}-c_{p0}\right)}{c_{p0}}$	*δ_P_* (15)	$P\;=\;\frac{\left(c_{p}-c_{p0}\right)}{\delta_{P}\cdot c_{p0}}\;N/{\mathrm{mm}}^{2}$ (10)	$\frac{\left(c_{p}-c_{p0}\right)}{c_{p0}}$	*δ_P_* (15)	$P\;=\;\frac{\left(c_{p}-c_{p0}\right)}{\delta_{P}\cdot c_{p0}}\;N/{\mathrm{mm}}^{2}$ (10)	$\frac{\left(c_{p}-c_{p0}\right)}{c_{p0}}$	*δ_P_* (15)	$P\;=\;\frac{\left(c_{p}-c_{p0}\right)}{\delta_{P}\cdot c_{p0}}\;N/{\mathrm{mm}}^{2}$ (10)
1	2	3	4	5	6	7	8	9	10	11
I-1	315	−0.0156	−0.0640	0.247	−0.0319	−0.0640	0.502	−0.0403	−0.0640	0.634
II-1	308	−0.0151	−0.0635	0.240	−0.0333	−0.0635	0.528	−0.0372	−0.0635	0.590
III-1	308	−0.0150	−0.0640	0.240	−0.0306	−0.0640	0.495	−0.0360	−0.0640	0.587

**Table 7 materials-14-03459-t007:** Results of calculations of normal stress *σ*_3_ in the wall using a limited number of measuring points.

Model	Number of Measurements *n*	0.25*σ*_3max_		0.50*σ*_3max_		0.75*σ*_3max_	
		$\frac{\left(c_{p}-c_{p0}\right)}{c_{p0}}$	$P\;=\;\frac{\left(c_{p}-c_{p0}\right)}{\delta_{P}\cdot c_{p0}}N/{\mathrm{mm}}^{2}$ (10)	$\frac{\left(c_{p}-c_{p0}\right)}{c_{p0}}$	$P\;=\;\frac{\left(c_{p}-c_{p0}\right)}{\delta_{P}\cdot c_{p0}}N/{\mathrm{mm}}^{2}$ (10)	$\frac{\left(c_{p}-c_{p0}\right)}{c_{p0}}$	$P\;=\;\frac{\left(c_{p}-c_{p0}\right)}{\delta_{P}\cdot c_{p0}}N/{\mathrm{mm}}^{2}$ (10)
**1**	**2**	**3**	**4**	**5**	**6**	**7**	**8**
I-1	45	−0.0115	0.181	−0.0261	0.410	−0.0337	0.528
II-1	44	−0.0104	0.166	−0.0293	0.464	−0.0316	0.501
III-1	44	−0.0119	0.187	−0.0270	0.424	−0.0324	0.509

**Table 8 materials-14-03459-t008:** Calculated mean stress *σ*_1_ based on a varying number of measuring points.

Model	Number of Measurements *n*	0.25*σ*_3max_			0.50*σ*_3max_			0.75*σ*_3max_		
		*σ*_3_, N/mm^2^ [7]	*P* N/mm^2^ (Table 5 and Table 6)	*σ*_1_, N/mm^2^ (20)	*σ*_3_, N/mm^2^ [7]	*P* N/mm^2^ (Table 5 and Table 6)	*σ*_1_, N/mm^2^ (20)	*σ*_3_, N/mm^2^ [7]	*P* N/mm^2^ (Table 5 and Table 6)	*σ*_1_, N/mm^2^ (20)
**1**	**2**	**3**	**4**	**5**	**6**	**7**	**8**	**9**	**10**	**11**
I-1	315	0.737	0.247	0.004	1.499	0.502	0.008	1.892	0.634	0.010
	45	0.540	0.181	0.003	1.224	0.410	0.007	1.577	0.528	0.008
II-1	308	0.714	0.240	0.005	1.573	0.528	0.011	1.757	0.590	0.012
	44	0.493	0.166	0.003	1.383	0.464	0.009	1.493	0.501	0.010
III-1	308	0.716	0.240	0.004	1.478	0.495	0.008	1.750	0.587	0.009
	44	0.557	0.187	0.003	1.265	0.424	0.007	1.518	0.509	0.008

**Table 9 materials-14-03459-t009:** FEM-based calculations for mean stresses *P* and *σ*_1_.

Model	Number of Measurements *n*	0.25*σ*_3max_			0.50*σ*_3max_			0.75*σ*_3max_		
		^FEM^*σ*_3_, N/mm^2^	^FEM^*P* N/mm^2^	^FEM^*σ*_1_, N/mm^2^	^FEM^*σ*_3_, N/mm^2^	^FEM^*P* N/mm^2^	^FEM^*σ*_1_, N/mm^2^	^FEM^*σ*_3_, N/mm^2^	^FEM^*P* N/mm^2^	^FEM^*σ*_1_, N/mm^2^
1	2	3	4	5	6	7	8	9	10	11
I-1	315	0.752	0.251	0.003	1.503	0.503	0.006	2.255	0.754	0.008
II-1	308	0.741	0.248	0.003	1.483	0.496	0.006	2.224	0.744	0.009
III-1	308	0.743	0.249	0.003	1.486	0.497	0.006	2.229	0.746	0.009

**Table 10 materials-14-03459-t010:** Compared mean values *P* and *σ* obtained from the tests and FEM calculations.

Model	Number of Measurements *n*	0.25*σ*_3max_			0.50*σ*_3max_			0.75*σ*_3max_		
		σ3σFEM3	PPFEM	σ1σFEM1	σ3σFEM3	PPFEM	σ1σFEM1	σ3σFEM3	PPFEM	σ1σFEM1
1	2	3	4	5	6	7	8	9	10	11
I-1	315	0.98	0.98	1.33	1.00	1.00	1.33	0.84	0.84	1.25
II-1	308	0.96	0.97	1.67	1.06	1.06	1.83	0.79	0.79	1.33
III-1	308	0.96	0.96	1.33	0.99	1.00	1.33	0.79	0.79	1.00
**on average:**		**0.97**	**0.97**	**1.44**	**1.02**	**1.02**	**1.50**	**0.81**	**0.81**	**1.19**
I-1	45	0.72	0.72	1.00	0.81	0.82	1.17	0.70	0.70	1.00
II-1	44	0.67	0.67	1.00	0.93	0.94	1.50	0.67	0.67	1.11
III-1	44	0.75	0.75	1.00	0.85	0.85	1.17	0.68	0.68	0.89
**on average:**		**0.71**	**0.71**	**1.00**	**0.87**	**0.87**	**1.28**	**0.68**	**0.69**	**1.00**

**Table 11 materials-14-03459-t011:** Minimum number of measuring points to determine the stresses *σ*_3_ and *P*.

Model	$\bar{x}=\sigma_{3\max}$ N/mm^2^	$\bar{x}=P$ N/mm^2^	$s_{\sigma 3}=\nu_{\mathrm{test}}\sigma_{3}{}_{\max}\;N/{\mathrm{mm}}^{2}$	$s_{P}=\nu_{\mathrm{test}}\sigma_{P}\;N/{\mathrm{mm}}^{2}$	*d_σ_*_3_ = 0.5α *σ*_3max_ N/mm^2^	*d_P_* = 0.5α *P* N/mm^2^	$\frac{t_{{}_{\alpha ,n-1}}^{2}s_{\sigma 3}^{2}}{d_{\sigma 3}^{2}}$	$\frac{t_{{}_{\alpha ,n-1}}^{2}s_{P}^{2}}{d_{P}^{2}}$
1	2	3	4	5	6	7	8	9
I-1	2.089	0.699	0.313	0.105	0.052	0.017	8	23
II-1	1.978	0.663	0.297	0.099	0.049	0.017	7	22
III-1	2.011	0.673	0.302	0.101	0.050	0.017	7	22

## Data Availability

Data available on request due to restrictions eg privacy or ethical The data presented in this study are available on request from the corresponding author. The data are not publicly available due to subsequent analyzes and publications.

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
