# Peer review of "Use of the AE Effect to Determine the Stresses State in AAC Masonry Walls under Compression"

_materials, 2021, doi:10.3390/ma14133459_

Round 1

Reviewer 1 Report

 The work elaborates a process for the non-destructive evaluation of autoclaved aerated concrete using ultrasonic techniques. The work contains both an extensive experimental and a numerical analysis components, while relevant theoretical aspects are presented. Quantitative measures on the correlation among the acoustoelastic coefficient and the density and moisture content of the AAC were obtained, while measuring specifications for the test repeatability and reliability were derived at the masonry scale. The work is overall of interest and within the scope of the journal. However, there are important methodological, analysis and coherence issues that need to be addressed. In particular:

  1. An elaborate explication of the intermediate steps relating Equations 1 and 2 with the resulting Equations 3 and 4 need to be provided. Several intermediate parameter definitions are missing (Such as P). Relevant analysis needs to be provided. An appendix reference form could be as well used.
  2. The same applies to the definitions of Eq. (11). The coefficient etaP used is not defned, while relevant explanations are missing. The same applies to the simplified forms of Eqs. (12) and (13). The definitions need to be self-contained and all relevant explanations need to appear for consistency and coherence.
  3. A physical/mechanical understanding needs to be provided for the substantial differences observed among numerical and experimental results in certain cases (50 and 44%). In other cases good agreement is however noted. What is the source of such discrepancies? Is the numerical model formulation applicable over the entire range stress states considered?
  4. The finite element results of Section 5 are given without any information on the models used. Relevant data need to be provided: convergence, type of elements, boundary conditions in the order they appear.
  5. The text is not properly related to the figures provided at several positions. Indicatively, section 4 and 5.
  6. The literature review should be extended to account for NDT and material anisotropy in the response of the base materials. Indicatively: 10.1098/rspa.2011.0666, 1016/j.matdes.2018.03.039
  7. The language of the manuscript needs to be thoroughly proof-read by an English language expert and several errors at different positions need to be corrected. Amongst others, verbs and articles are missing or erroneously used in different sentences.

Author Response

Thank you very much for the detailed review. The answers are included in the attached file. 

Reviewer 2 Report

The work is very interesting. The results are clearly explained. I congratulate the authors for the way they made and presented this work. 

Author Response

Thank you very much for the positive reception of our article. We have included the acknowledgment in the file. 

Reviewer 3 Report

The paper presents the experimental validation of the theory (related to AE) previously developed by the authors. The experiments were extensive and results clearly present. There are a lot of improvements to be made to the paper in terms of its readability. 

  1. The abstract is missing.
  2. Lots of spelling mistakes in the text and references. For example: Ref 21: Murnagham?
  3. Lot of parenthesis and brackets aren't closed.
  4. The variables in the equation when present in the text should be italicized. For example 1 and l look the same.
  5. Eq. 10 is not clear. Please elaborate the steps.
  6. What is the significance of negative relative coefficient?
  7. Labels in the figures are cut off. Lots of figures are small and are not clear. Please redo them.
  8. Page 24: itemized numbering should start from 1.
  9. Page 24: reference for the minimum number of samples should be provided.
  10. Usually Bayesian methods (Structural Safety 40 (2013): 21-30) are used to update/validate the FE models. Perhaps the authors could briefly discuss why these weren't considered, since the global objective is to assess the safety and reliability of used constructions.

Author Response

(The authors gave the same response as above.)

Round 2

Reviewer 1 Report

The authors have addressed all issues raised and accordingly revised the manuscript. I suggest that the work is published after minor text editing.